# Multiple Physical Quantities Janus Metastructure Sensor Based on PSHE

**DOI:** 10.3390/s23104747

**Published:** 2023-05-14

**Authors:** Junyang Sui, Jie Xu, Aowei Liang, Jiahao Zou, Chuanqi Wu, Tinghao Zhang, Haifeng Zhang

**Affiliations:** College of Electronic and Optical Engineering & College of Flexible Electronics (Future Technology), Nanjing University of Posts and Telecommunications, Nanjing 210023, China; b20050530@njupt.edu.cn (J.S.); b21021927@njupt.edu.cn (J.X.); b20021631@njupt.edu.cn (A.L.); b22021216@njupt.edu.cn (J.Z.); b22021517@njupt.edu.cn (C.W.); b22021319@njupt.edu.cn (T.Z.)

**Keywords:** metastructure, Janus metastructure, photonic spin Hall effect, physical quantities detection

## Abstract

In this paper, a Janus metastructure sensor (JMS) based on the photonic spin Hall effect (PSHE), which can detect multiple physical quantities, is proposed. The Janus property is derived from the fact that the asymmetric arrangement of different dielectrics breaks the structure parity. Hence, the metastructure is endowed with different detection performances for physical quantities on multiple scales, broadening the range and improving the accuracy of the detection. When electromagnetic waves (EWs) are incident from the forward scale of the JMS, the refractive index, thickness, and incidence angle can be detected by locking the angle corresponding to the PSHE displacement peak that is enhanced by the graphene. The relevant detection ranges are 2~2.4, 2~2.35 μm, and 27°~47°, with sensitivities (S) of 81.35°/RIU, 64.84°/μm, and 0.02238 THz/°, respectively. Under the condition that EWs incident into the JMS from the backward direction, the JMS can also detect the same physical quantities with different sensing properties, such as S of 99.3°/RIU, 70.07°/μm, and 0.02348 THz/° in corresponding detection ranges of 2~2.09, 1.85~2.02 μm, and 20°~40°. This novel multifunctional JMS is a supplement to the traditional single-function sensor and has a certain prospect in the field of multiscenario applications.

## 1. Introduction

Janus is the god of creation in Roman mythology, with two different faces, one facing the past and the other facing the future [1]. Inspired by this, researchers first named two-sided particles with different materials on opposite sides the Janus particles [2,3,4]. Due to the difference of materials on both sides, electromagnetic waves (EWs) can show different electromagnetic characteristics when incident into Janus particles from different directions, which adds the regulatory features of EWs in the direction dimension [5,6,7,8]. Recently, similar Janus functions have been extended to the field of the metastructure, which refers to quasi-periodic structures formed by artificially constructed dielectrics. The metastructure has physical properties not found in natural structures and can modulate the amplitude, phase, polarization, and angular momentum of EWs [9,10,11]. By using special methods, such as asymmetric arrangement to break the parity of structures [12], the metastructure is equipped with the Janus property and can regulate the electromagnetic performance of EWs propagating in different directions, which greatly expands the application field of the metastructure. Chen et al. [13] proposed a Janus metastructure (JMS) with transmission reflection on the forward scale and polarization regulation on the backward scale, which effectively compressed the working distance of the imaging system by using its asymmetric propagation characteristics and also overcame the energy dispersion problem of the traditional pancake system. Yang et al. [14] designed and prepared a kind of JMS with cooling and heating dual functions. Its infrared emissivity was as high as 97.2%, which could be used for zero-energy thermal management throughout the year.

The photonic spin Hall effect (PSHE) [15] refers to when the process of total reflection, affected by the conservation of spin angular momentum and orbital angular momentum, the left and right circularly polarized components of linearly polarized light are split in opposite transverse directions perpendicular to the gradient of refractive index (RI) according to the direction of rotation, resulting in two beams of light. The spin offset in PSHE is very sensitive to the change of physical parameters of the system, so it has great application potential in precise measurement [16]. With the weak measurement technique proposed [17], the displacement phenomenon of PSHE is amplified 10^4^ times, which is convenient for experimental observation. Researchers have also found that the PSHE phenomenon can be effectively enhanced by introducing a graphene layer and tuning its chemical potential [18]. These findings provide ideas for improving PSHE in the terahertz (THz) range.

With the rapid development of sensor research, it has become one of the three key technologies in the information industry together with computer technology and information technology [19]. The optical sensor based on a THz band is considered a promising physical quantity detection tool due to the advantages of having high sensitivity (S), having no label, being nondestructive, and having real-time monitoring [20]. Therefore, it is widely used in biomedical [21], nondestructive testing [22], and other fields. In recent years, Cheng et al. [23] proposed a novel THz RI sensor based on PSHE for cancer detection, which exhibits S of 6.1 × 10^5^ μm/RIU under optimal pumping power and could distinguish normal gastric cells and corresponding cancer cells. Zhu et al. [24] designed a Tamm structure, which was able to achieve RI detection with S = 2804 mm/RIU in a THz band with a resolution of up to 10^−8^ RIU by using PSHE. Kumar et al. [25] reported a PSHE plasma sensor based on a graphene monolayer under a THz environment. It could realize the gas sensor and the detection limit was up to 10^−5^ RIU, which could be useful for the early detection of airborne viruses such as SARS-CoV-2. All of the above reports could realize the detection of the physical quantity in the THz range through PSHE and could have an excellent sensing performance, but, unfortunately, the realized functions are single. Liu et al. [26] proposed a PSHE sensor for high-precision RI detection and graphene layers’ number detection. By locking the corresponding angle of the PSHE peak, the sensor could detect the RI of S = 127.85°/RIU and the 1~9 layers’ number of graphene layers with S = 4.54°/layer. The multifunctional sensor provided a new idea for the research of related fields and exhibited certain research values.

In this paper, a JMS based on PSHE is proposed that can realize the multiscale multiple physical quantities detection in the THz band, making up for the deficiency of the traditional sensor single function. The PSHE phenomenon is enhanced by selecting the appropriate chemical potential of graphene layers. By locking the EWs incidence angle corresponding to the PSHE displacement peak, the JMS can simultaneously detect RI, thickness, and angle on both forward and backward scales with different performance (as shown in Table 1). When EWs incident into the JMS from the front, RI, thickness, and angle in the range of 2~2.4, 2~2.35 μm, and 27°~47° can be detected; the S corresponds to 81.35°/RIU, 64.84°/μm, and 0.02238 THz/°, respectively. Under the condition of the backward propagation of EWs, the detection ranges of RI, thickness, and angle are 2~2.09, 1.85~2.02 μm, and 20°~40° with S of 99.3°/RIU, 70.07°/μm, and 0.02348 THz/°. It is not difficult to find that the multiscale physical quantity measurement makes the detection range of the JMS larger. In addition, the same physical quantity has a small common detection range in the forward and backward detection, but the corresponding detection performance is different. Therefore, the physical quantity of a certain value can be compared for forward and backward detection to verify whether the detection result is correct, which improves the accuracy of the JMS detection. It is proposed that the multiscale and multifunction JMS with high S, no label, no damage, and real-time monitoring can be applied to a variety of application scenarios and is able to ensure the accuracy of its detection, which has certain research value.

## 2. The Theoretical Model

Figure 1 shows the structure belonging to the JMS, which can be fabricated by etching [27]. In order to adapt to the common condition, the JMS is exposed to air and operates at *T* = 300 K. The red and blue columns indicate that the EWs incident forward and backward, respectively, at an angle *θ* to the *z*-axis. Figure 1 also exhibits the setting of a Gaussian beam incident at a certain angle spectrum at the first dielectric surface, using green and yellow beams to separately represent the left-handed circularly polarized component *δ^H^*_−_ and right-handed circularly polarized component *δ^H^*_−_. The RI of dielectrics A and B are *n*_A_ = 1.7 and *n*_B_ = 2, respectively. It should be emphasized that Leiwin et al. [28] derived the expression of effective permittivity and permeability of composite materials based on the Mie resonance theory and that the required RI could be obtained in a wide range. This technology has been applied in practice [29], so the dielectric RI set in this JMS is reasonable and available. According to the Herzberger equation, in the THz band, RI of Si is considered to be *n*_Si_ = 3.419 [30]. The electric field conductivity σ of graphene is composed of intraband *σ*_intra_ and interband *σ*_inter_ [31].
(1)$$\sigma =\frac{ie^{2}k_{B}T}{\pi \hbar^{2}(\omega +i/\tau )}(\frac{\mu_{C}}{k_{B}T}+2\ln(e^{-\frac{\mu_{C}}{k_{B}T}}+1))+\frac{ie^{2}}{4\pi \hbar }\ln\left|\frac{2\mu_{C}-\hbar (\omega +i/\tau )}{2\mu_{C}+\hbar (\omega +i/\tau )}\right|,$$
where *ω*, *k*_B_, ħ, *e*, *T*, *μ*_C_, and *τ* represent the angular frequency, Boltzmann’s constant, Planck’s constant, electron charge, temperature, chemical potential, and carrier relaxation time, respectively. There is a specific functional relationship between the conductivity and the chemical potential of graphene, which is different from that of ordinary dielectric. Assuming that the electronic energy band of a graphene layer is not affected by adjacent elements, the effective dielectric constant *ε*_G_ of graphene can be written as [31]:(2)$$\varepsilon_{G}=1+\frac{i\sigma }{\omega \varepsilon_{0}d_{GL}},$$
where *ε*_0_ is the vacuum dielectric constant. So, the RI of graphene layer is written as *n*_G_ = (*ε*_G_)^1/2^. For the ordinary dielectric and graphene layers, their transfer matrix can be expressed as [32]:(3)$$M_{i}=\left(\begin{array}{cc} \cos(k_{iz}d_{i}) & -\frac{i}{\eta_{i}}\sin(k_{iz}d_{i})\\ -i\eta_{i}\sin(k_{iz}d_{i}) & \cos(k_{iz}d_{i}) \end{array}\right),$$
where *i* can be represented by A, B, Si, and graphene, symbolizing the transmission matrix of different ordinary dielectrics. *k_jz_* = *ω*/c*n_i_*sin*θ_i_* is the component of the wave vector on the *z*-axis; the speed of light in a vacuum is symbolized by *c*. The definition of *s*-wave and *p*-wave can be referred to Ref. [33]. *η_i_* is the light conductivity; for *s*-wave, *η_i_* = (*ε*_0_/*μ*_0_)^1/2^*n_i_*cos*θ_i_*. For *p*-wave, then *η_i_* = (*ε*_0_/*μ*_0_)^1/2^*n_i_*/cos*θ_i_*. *ε*_0_ and *μ*_0_ are vacuum dielectric constants and permeability, respectively. The transmission matrix of (AB)^6^(GSi)^3^(AB)^4^ is [32]:(4)$$M=\sum_{i}^{26}M_{i}=\left(\begin{array}{cc} m_{11} & m_{12}\\ m_{21} & m_{22} \end{array}\right).$$

The reflection and transmission coefficients symbolized by *r* and *t* can be expressed as [32]:(5)$$r=\frac{\left(m_{11}+m_{12}\eta_{0})\eta_{0}-(m_{21}+m_{22}\eta_{0}\right)}{\left(m_{11}+m_{12}\eta_{0})\eta_{0}+(m_{21}+m_{22}\eta_{0}\right)}.$$
(6)$$t=\frac{2\eta_{0}}{\left(m_{11}+m_{12}\eta_{0})\eta_{0}+(m_{21}+m_{22}\eta_{0}\right)}.$$

The *R* = |*r*|^2^ and *T* = |*t*|^2^ separately represent reflectance (*R*) and transmittance (*T*). The absorptance (*A*) is written through [32]:(7)A=1−R−T.

Gaussian beams with a certain angle spectrum can be expressed as [34]:(8)$$\tilde{E}i\pm =(eix+ioeiy)\frac{\omega_{0}}{\sqrt{2\pi }}\exp[-\frac{\omega_{0}^{2}(k_{ix}^{2}+k_{iy}^{2})}{4}],$$
where *ω*_0_ represents the beam waist and *o* is the polarization operator. Left-handed and right-handed circular polarized beams are represented by *o* = 1 and *o* = −1, respectively. The horizontal and vertical polarization states are separately symbolized by *H* and *V*. A matrix of coefficients between an incident and reflected electric fields can be expressed as [34]:(9)$$\left[\begin{array}{c} {\tilde{E}}_{r}^{H}\\ {\tilde{E}}_{r}^{V} \end{array}\right]=\left[\begin{array}{cc} r^{p} & \frac{kry\cot\theta_{i}(r^{p}+r^{s})}{k_{0}}\\ -\frac{kry\cot\theta_{i}(r^{p}+r^{s})}{k_{0}} & r^{s} \end{array}\right]\left[\begin{array}{c} {\tilde{E}}_{i}^{H}\\ {\tilde{E}}_{i}^{V} \end{array}\right],$$
*k*_0_ symbolizes the number of waves in free space. *r^p^* and *r^s^* represent the Fresnel reflection coefficients of the *p*-wave and *s*-wave, respectively. According to Equations (8) and (9), the expression of the spectrum of the reflection angle can be obtained [34]:(10)$$\begin{array}{l} {\tilde{E}}_{r}^{H}=\frac{r_{p}}{\sqrt{2}}[\exp(+ik_{ry}\delta_{r}^{H}){\tilde{E}}_{r+}+\exp(-ik_{ry}\delta_{r}^{H}){\tilde{E}}_{r-}],\\ {\tilde{E}}_{r}^{V}=\frac{ir_{s}}{\sqrt{2}}[-\exp(+ik_{ry}\delta_{r}^{V}){\tilde{E}}_{r+}+\exp(-ik_{ry}\delta_{r}^{V}){\tilde{E}}_{r-}]. \end{array}$$

Here *δ^H^_r_* = (1 + *r^s^/r^p^*)cot*θ_i_*/*k*_0_ and *δ^V^_r_* = (1 + *r^p^/r^s^*)cot*θ_i_*/*k*_0_. ${\tilde{E}}_{r\pm }$ can be written in a similar style to Equation (8). $\varphi^{s}$ and $\varphi^{p}$ symbolize the phase of *r*^s^ and *r^p^*. For the reflected light, the PSHE lateral displacement of the left-handed and right-handed components can be expressed as [34]:(11)$$\begin{array}{l} \delta_{\pm }^{H}=\mp \frac{\lambda }{2\pi }[1+\frac{\left|r^{s}\right|}{\left|r^{p}\right|}\cos(\varphi^{s}-\varphi^{p})]\cot\theta_{i},\\ \delta_{\pm }^{V}=\mp \frac{\lambda }{2\pi }[1+\frac{\left|r^{p}\right|}{\left|r^{s}\right|}\cos(\varphi^{p}-\varphi^{s})]\cot\theta_{i}. \end{array}$$

In this paper, we only discuss the case of left-handed circularly polarized component displacement *δ^H^*_−_.

## 3. Analysis and Discussion of Performances

By varying the external voltage, the *μ*_C_ of the graphene can be adjusted [31]. How to change the graphene layer *μ*_C,_ refer to Ref. [35]. In order to explain the generation of *δ^H^*_−_ peak and the choice of *μ*_C_, taking the EWs propagation from the forward direction at the frequency of 5.52 THz to detect the RI of dielectric B *n*_B_ as an example. Figure 2 displays the real and imaginary parts of the graphene surface conductivity *σ* at different *μ*_C_. According to Equation (1), *μ*_C_ affects the *σ* and the *σ* increases with the rise of *μ*_C_. According to Equation (2), the change of *σ* will further change the permittivity of graphene layers, which are at different positions in the structure, thus affecting the effective permittivity and impedance of the whole structure. As a result, when EWs propagate through the structure, the electromagnetic properties such as reflection coefficient will be changed. Here, it takes the four classical *μ*_C_ of 0.2 eV, 0.4 eV, 0.6 eV, and 0.8 eV. Figure 3 shows the relationship between the absolute values |*r^s^*| and |*r^p^*| of Fresnel coefficients and the *θ* at different *μ*_C_. The solid yellow and dashed green lines severally symbolize the reflection coefficient curves of |*r^s^*| and |*r^p^*|. The variation of *μ*_C_ will affect the *σ*, thus altering the Fresnel coefficients and regulating *δ^H^*_−_. Moreover, the energy is localized and the reflection gap is created, where |*r^s^*| and |*r^p^*| drop quickly to produce defect peaks as a result of the introduction of the defect layer. Under various *μ*_C_, the reflection gap is produced at different *θ*. As can be seen from Figure 3a–d, the *θ* corresponding to the curve peaks of |*r^s^*| and |*r^p^*| gradually become smaller. When *μ*_C_ = 0.6 eV, the peak value of |*r^p^*| reaches the minimum at 18.67°, where the defect peak generates |*r^p^*| = 0.002. By the beam displacement of Equation (11), the division of the spin correlation primarily depends on the part of |*r^s^*|/|*r^p^*|, thus the |*r^s^*|/|*r^p^*| might reach a high value close to the defect peak of *r^p^*|, resulting in the peak of *δ^H^*_−_. In Figure 4, this theory is put to the test. Figure 4a,b displays the *δ^H^*_−_ values at various *μ*_C_ and, as *μ*_C_ rises, the *δ^H^*_−_ peak progressively shifts to a small angle. *δ^H^*_−_ produces the highest peak at 18.76°; *δ^H^*_−_ = 2.46 × 10^−4^ m when *μ*_C_ = 0.6 eV. *δ^H^*_−_ peaks at *μ*_C_ of 0.2 eV, 0.4 eV, and 0.8 eV are small, the values are 2.94 × 10^−6^ m, −1.02 × 10^−6^ m, and −1.14 × 10^−6^ m, belonging to *θ* of 19.48°, 29.7°, and 33.07°, respectively. To choose the suitable *μ*_C_ with greater certainty, Figure 3c shows the changing pattern of the *δ^H^*_−_ peak values corresponding to different *μ*_C_ within the *n*_B_ range of 2~2.4. It is evident that the peak value of *δ^H^*_−_ at *μ*_C_ = 0.6 eV is substantially higher than values at other *μ*_C_ and that it varies greatly with the RI. The choice of *μ*_C_ = 0.6 eV has great sensing performance because the multiple physical quantities detection is accomplished by locking the *δ^H^*_−_ peak.

Similar to how changes in wave vectors and phases are influenced by changes in *μ*_C_, RI modulation will have an impact on the size of the Fresnel reflection coefficients |*r^s^*| and |*r^p^*|. As a result, both *δ^H^*_−_ peak and *θ* vary accordingly. So, RI detection can be accomplished by locking the corresponding *θ* of the *δ^H^*_−_ peak. The dielectric B layers are selected as the detection region. When EWs propagate forward at 5.52 THz, Figure 5a indicates that continuous *θ* of the *δ^H^*_−_ peak exhibits a good linear fitting relationship (LFR) in the range of *n*_B_ from 2 to 2.4. In this scope, the values of *δ^H^*_−_ are greater than 6.89 × 10^−5^ m, which can ensure basic detectability. Using the linear fitting method, equidistant locations along the horizontal axis are chosen in order to produce the LFR. Figure 5b exhibits the LFR between *n*_B_ and *θ*. In the range of RI of 2~2.4, the LFR is *θ* = 81.35 *n*_B_ − 142.4. R^2^ is applied to evaluate the quality of linear fit. R^2^ = 0.9928 proves that the sensor is reliable and S can reach 81.38°/RIU. Figure 6 displays the RI detection performance under the condition of EWs backward propagation at *f* = 5.62 THz. As exhibited in Figure 6a, with the increase in *n*_B_ from 2 to 2.09, the *θ* of the *δ^H^*_−_ peak exhibits linear change along with *δ^H^*_−_ > 5.87 × 10^−5^ m. Figure 6b demonstrates the LFR between *n*_B_ and *θ*. Between RI from 2 to 2.09, the LFR is *θ* = 99.3*n*_B_−156.3 and the S is up to 99.3 °/RIU. R^2^ = 0.9928 indicates that the detection is reliable. Because EWs incident forward and backward separately have different RI detection performance in the common range of 2~2.09, an unknown RI is detected simultaneously. The *δ^H^*_−_ peak is examined to have a maximum at *θ* = 20.3° on the forward scale and a maximum at *θ* = 42.3° on the backward scale. Through the corresponding forward and backward LFR, the unknown RI can be obtained as 2, which can mutually verify the accuracy of the test results.

The precise measurement of thin film thickness has important application in industrial production [36]. The JMS proposed can realize the micron thickness change detection by locking the *θ* of the *δ^H^*_−_ peak. On the forward scale, by investigating the changes in *θ* with *d*B from 2 to 2.35 μm, the relationship between *θ* and *d*B is established and depicted in Figure 7. Figure 7a displays the phenomenon of continuous variation of *δ^H^*_−_ in the range of *d*B = 2~2.35 μm; *δ^H^*_−_ is greater than 2.18 × 10^−4^ m. The results show a linear distribution in a certain measurement range and, by further exploring the relationship between the two physical quantities, a fitting curve of *θ* and *d*B is obtained. The LFR is *θ* = 64.84 *d*B−109.6. The R^2^ is found to be high, at 0.9904. S, an important indicator of sensor performance, is measured to be as high as 64.68 °/μm, indicating the high performance of the sensor manufacturing.

When EWs are incident backward, the detection range of *d*B is 1.85~2.02 μm. The difference between forward and backward detection is of practical significance to sensor applications as the wider measurement range expands the working range of the thickness sensor. As shown in Figure 8a, the *δ^H^*_−_ peak values are greater than 2.18 × 10^−4^ m in the *d*B scope of 1.85~2.02 μm. Figure 8b selects six data points at the same intervals for linear fitting to verify the strong LFR. The results exhibit that, as the thickness varies from 1.85 to 2.02 μm, the LFR is *θ* = 70.07 *d*B−102.3, with an R^2^ of 0.9993. The S is 70.07°/μm, indicating further possibilities for sensor fabrication.

The proposed JMS can also detect the *θ* of EWs within a certain incidence frequency range. When EWs propagate in the forward and backward directions, Figure 9a and Figure 10a show a good linear relationship between *θ* and *f* via the *δ^H^*_−_ peak. As the angle *θ* increases in the range of 27°~47° and 20°~40°, the *δ^H^*_−_ peak produces a blue shift, respectively. Meanwhile, the corresponding *δ^H^*_−_ peaks still remain larger than 9.93 × 10^−5^ m and 1.05 × 10^−6^ m, proving basic detectability. With the purpose of exploring their linear relationship, the correlative LFRs *δ^H^*_−_ = 0.02238*θ* + 4.911 and *δ^H^*_−_ = 0.02348*θ* + 7.64 are presented in Figure 9b and Figure 10b; 0.02238 THz/° and 0.02348 THz/° are the S compared with the magnetized plasma angle sensor with S up to 1.325 × 10^−4^ THz/° [37]. The JMS is more responsive to the changes in *θ*. R^2^ are all 0.99, indicating the high quality of the LFR. The above research presents that the sensor exhibits multiscale different *θ* detection ranges and the LFRs of detection are excellent, with sensitive response and exact detection, thus providing a novel and stable way to detect weak *θ* change in the THz band.

Finally, to more intuitively describe the advantages of the JMS, Table 2 summarizes the sensors with excellent performance reported in the past and compares them with the JMS designed in this work. From the evaluation aspects of Janus and multifunction and physical quantities detection performance, the JMS proposed in this paper has a more advanced application value.

## 4. Conclusions

To sum up, this paper theoretically studies multiple physical quantities of JMS based on the PSHE. Through the asymmetric arrangement of different dielectrics, the parity of the metastructure is broken and the metastructure is endowed with Janus property to realize multiscale physical quantities detection. Graphene is introduced into the structure to enhance the PSHE through its tunable chemical potential. By locking the *θ* corresponding to the PSHE displacement peak, the JMS can detect RI, thickness, and angle on both forward and backward scales, exhibiting different detection performance, which not only broadens the detection range but also improves the detection accuracy. For RI detection, the JMS detection ranges are 2~2.4 and 2~2.09 on the forward and backward scales, respectively, with S of 81.35°/RIU and 99.3°/RIU. The JMS can detect the thickness ranges of 2~2.35 μm and 1.85~2.02 μm on the opposite scales, with S of 64.8°/μm and 70.07°/μm. Additionally, angles in the range of 27~47 and 20~40 can also be detected by the JMS, with S of 0.02238 THz/° and 0.02348 THz/°. The proposed multiscale and multifunctional JMS has the advantages of high S, no label, no damage, and real-time monitoring, which can be applied to various application scenarios and can provide a new way for the design of novel multifunction devices.

## Figures and Tables

**Figure 1 sensors-23-04747-f001:**
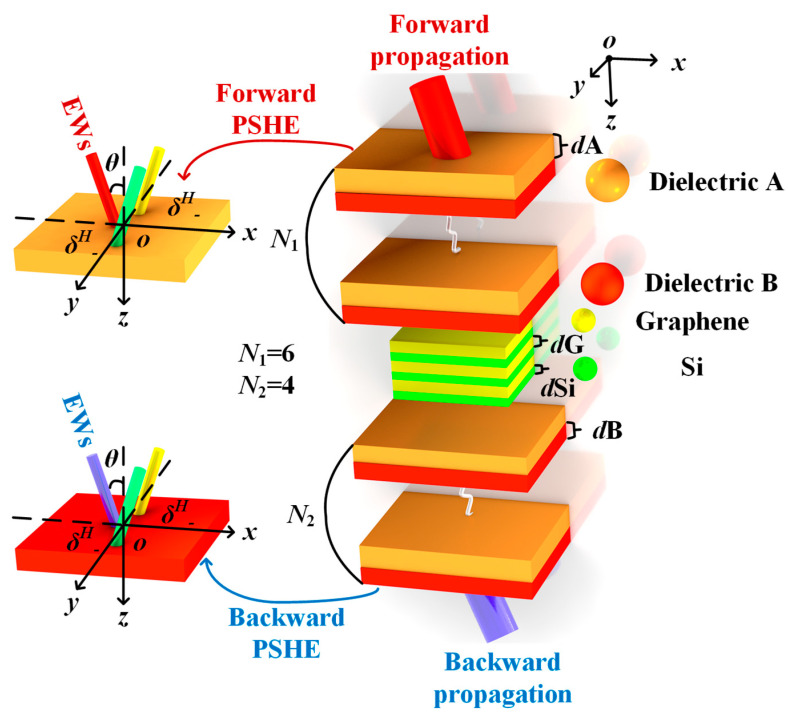
The structure diagram of the JMS is arranged asymmetrically by graphene layer and common dielectrics are filled with different colors. The entire structure is (AB)*^N^*^1^(GSi)^3^(AB)*^N^*^2^, where *N*_1_ = 6 and *N*_2_ = 4. The thickness of the dielectric A, dielectric B, Si, and graphene are *d*A = 4 μm, *d*B = 2 μm, *d*Si = 1 μm, and *d*G = 0.34 nm, respectively.

**Figure 2 sensors-23-04747-f002:**
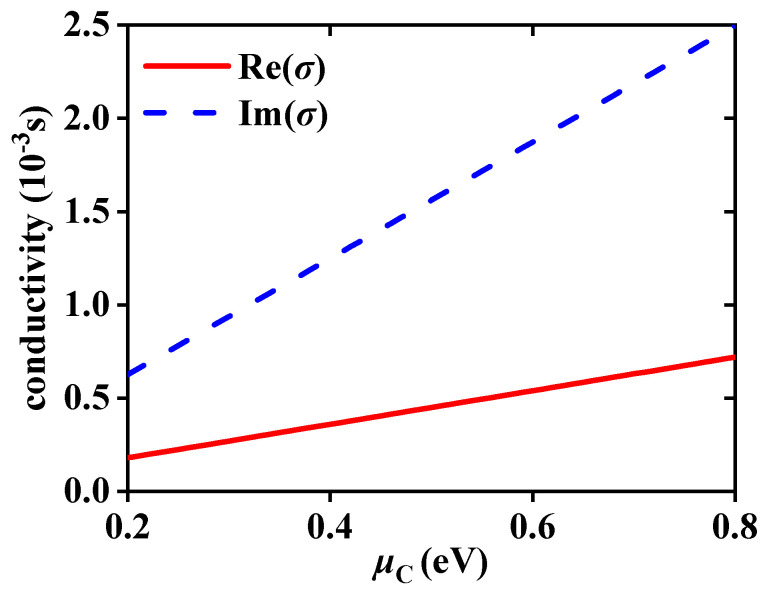
The real and imaginary parts of the graphene surface conductivity under different *μ*_C_.

**Figure 3 sensors-23-04747-f003:**
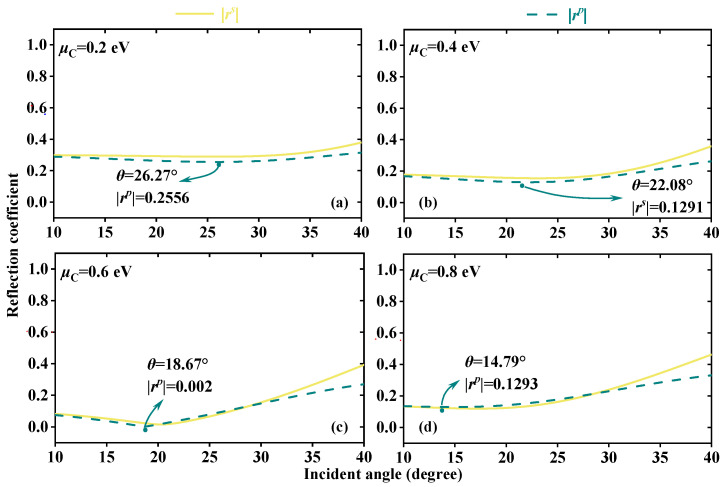
The reflection coefficient curves of |*r^s^*| and |*r^p^*| with different *μ*_C_; (**a**) *μ*_C_ = 0.2 eV, (**b**) *μ*_C_ = 0.4 eV, (**c**) *μ*_C_ = 0.6 eV, (**d**) *μ*_C_ = 0.8 eV.

**Figure 4 sensors-23-04747-f004:**
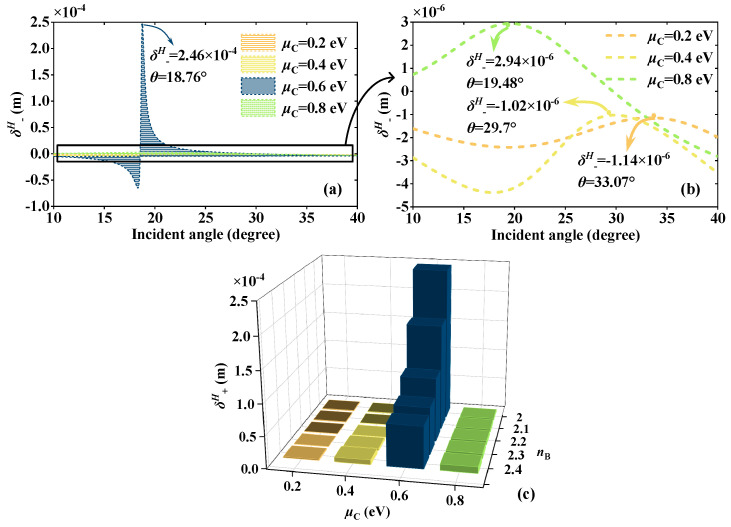
When *μ*_C_ changes and EWs are incident from the front; (**a**,**b**) the comparison plots of *δ^H^*_−_ under *n*_B_ = 2. (**c**) Plots of *δ^H^*_−_ peak values under different *n*_B_.

**Figure 5 sensors-23-04747-f005:**
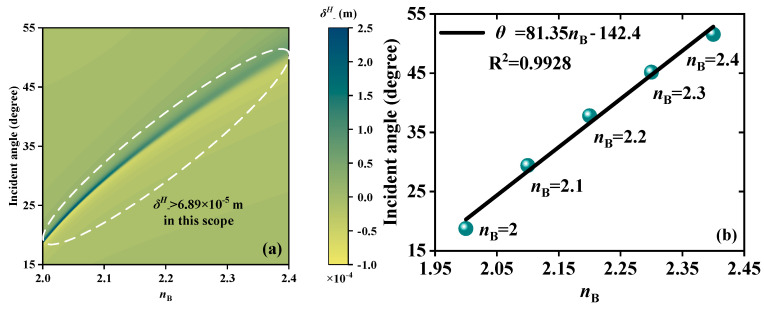
Schematic diagrams of the RI detection when EWs propagate forward; the detection scope is *n*_B_ from 2 to 2.4. (**a**) Continuous varying *δ^H^*_−_ peaks. (**b**) The LFR between *n*_B_ and *θ*; the LFR is *θ* = 81.35 *n*_B_–142.4.

**Figure 6 sensors-23-04747-f006:**
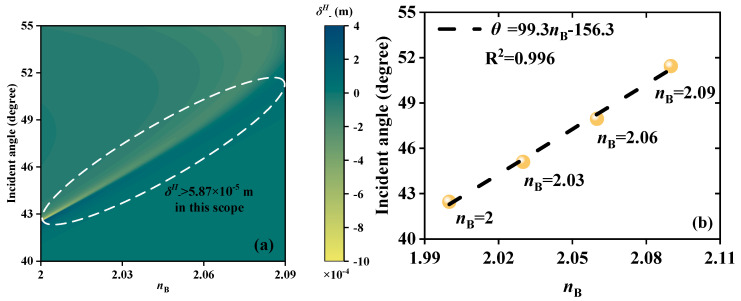
Schematic diagrams of the RI detection when EWs propagate backward; the detection scope is *n*_B_ from 2 to 2.09. (**a**) Continuous varying *δ^H^*_−_ peaks. (**b**) The LFR between *n*_B_ and *θ*; the LFR is *δ^H^*_−_ = 99.3 *n*_B_ − 156.3.

**Figure 7 sensors-23-04747-f007:**
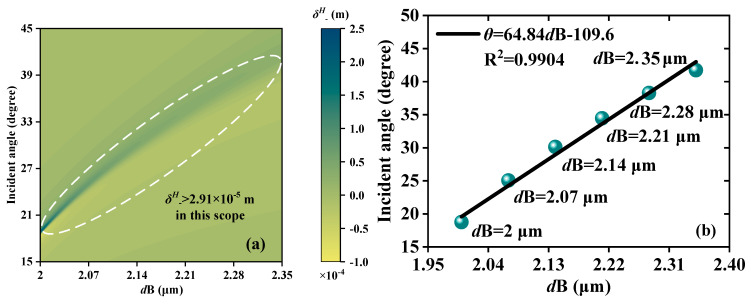
Schematic diagrams of the thickness detection when EWs propagate forward; the detection scope is *d*B from 2 μm to 2.35 μm. (**a**) Continuous varying *δ^H^*_−_ peaks. (**b**) The LFR between *d*B and *θ*; the LFR is *θ* = 64.84 *d*B−109.6.

**Figure 8 sensors-23-04747-f008:**
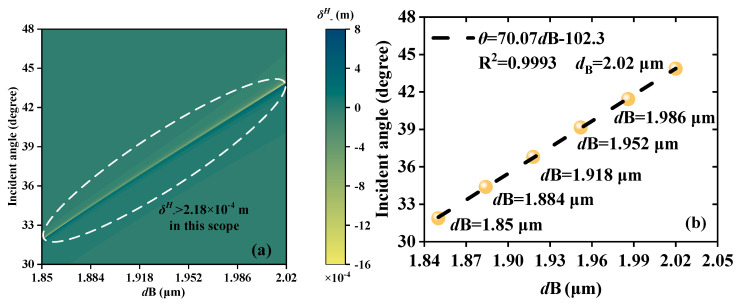
Schematic diagrams of the thickness detection when EWs propagate backward; the detection scope is *d*B from 1.85 μm to 2.02 μm. (**a**) Continuous varying *δ^H^*_−_ peaks. (**b**) The LFR between *d*B and *θ*; the LFR is *θ* = 70.07 *d*B−102.3.

**Figure 9 sensors-23-04747-f009:**
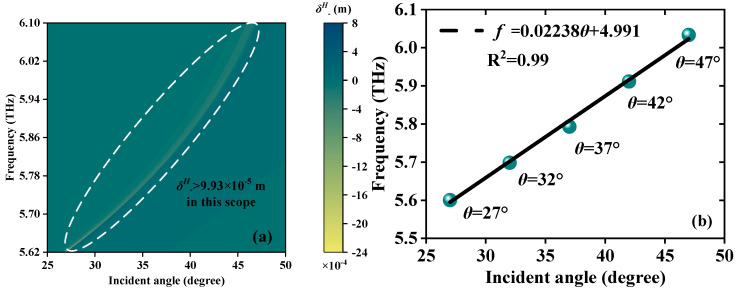
Schematic diagrams of the angle detection when EWs propagate forward; the detection scope is *θ* from 27° to 47°. (**a**) Continuous varying *δ^H^*_−_ peaks. (**b**) The LFR between *θ* and frequency; the LFR is *f* = 0.02238*θ* + 4.991.

**Figure 10 sensors-23-04747-f010:**
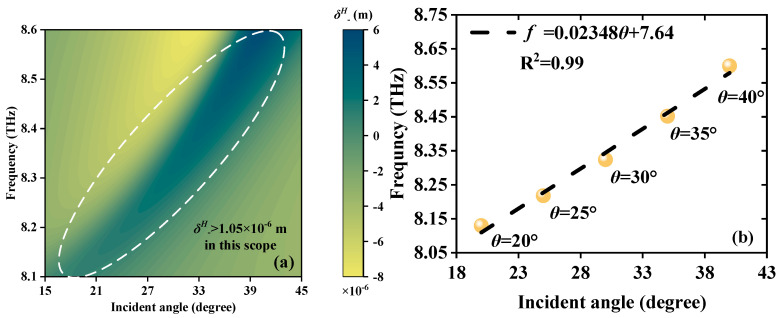
Schematic diagrams of the angle detection when EWs propagate backward; the detection scope is *θ* from 20° to 40°. (**a**) Continuous varying *δ^H^*_−_ peaks. (**b**) The LFR between *θ* and frequency; the LFR is *f* = 0.02348*θ* + 7.64.

**Table 1 sensors-23-04747-t001:** The Janus performance of the JMS.

		RI	Thickness (μm)	Angle (°)
Forward	Range	2~2.4	2~2.35	27~47
	S	81.35 °/RIU	64.84 °/μm	0.02238 THz/°
Backward	Range	2~2.09	1.85~2.02	20~40
	S	99.3 °/RIU	70.07 °/μm	0.02348 THz/°

**Table 2 sensors-23-04747-t002:** The performance of the traditional sensors compared with the proposed JMS.

Refs.	Janus	Multifunction	Physical Quantities Detection			
[38]	No	No	RI	Range		1.362~1.366
				S		303,376 nm/RIU
[39]	No	No	Thickness	Range		0~0.5 μm
				S		/
[40]	No	No	Angle	Range		0~45
				S		55.67 pm/°
[37]	Yes	No	RI	Forward	Range	1.35~2.09
					S	132 MHz/RIU
				Backward	Range	1~1.57
					S	40.7 MHz/RIU
[41]	No	Yes	RI	Range		2~2.7
				S		32.3 THz/RIU
			Angle	Range		25°~70°
				S		0.5 THz/°
This work	Yes	Yes	RI	Forward	Range	2~2.4
					S	81.35°/RIU
				Backward	Range	2~2.09
					S	99.3°/RIU
			Thickness	Indicated in the article		
			Angle	Indicated in the article		

## Data Availability

Samples of the compounds are available from the authors.

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
