# Peer review of "Multiple Physical Quantities Janus Metastructure Sensor Based on PSHE"

_sensors, 2023, doi:10.3390/s23104747_

Round 1

Reviewer 1 Report

In this work, the authors simulate the operation of a multi-sensor based on the Janus metastructure. The multilayer structure itself (Fig. 1) is borrowed from [31–33]. The equations describing the passage of light through the layers of the structure are borrowed from [32]. The equations describing the shift of the left and right circular polarization upon reflection from such a structure are borrowed from [34]. The authors only need to substitute the layer parameters into the equations and calculate the PSHE. The simulation results are new and the work can be published after the authors take into account the comments.

Comments

1. The paper lists the obtained sensitivities for measuring the refractive index, thickness, and angle of incidence several times. To these data in the Abstract should be added the measurement range of these quantities: RI 2-2.4, db 2-2.35 um, angle of incidence 27-47 degrees.

2. The paper measures the refractive index and the thickness of one layer from the structure itself (nB, dB). But there are several such layers in the structure (N2=4). It is necessary to explain to the reader why the sensor measures not the thickness and refractive index of the outer film, but its own layers. It turns out that in order to measure RI and layer thickness with this sensor in practice, one must first make a metastructure of 26 layers (Fig. 1). The authors should explain to the reader why the sensor proposed in the paper measures its own parameters.

3. In the paper, the selected thicknesses of all different layers of the structure in Fig. 1 should be indicated: dA, dGL, dSi, dB.

4. In eq. (4) matrix elements are denoted as m, and in (5) they are also denoted by M.

Author Response

The details can be seen the attachment, which is a response letter to the comments of Reviewer

Reviewer 2 Report

This manuscript demonstrates a Janus metastructure sensor (JMS) based on photonic Spin Hall effect (PSHE) which can realize multiple physical quantities detection in the THz band. Overall, the manuscript is well-structured and provides a clear description of the device. The authors have well explained the Janus property, which is derived from the asymmetric arrangement of different dielectrics that breaks the parity of the structure. The use of graphene to enhance the PSHE displacement peak is also well demonstrated. The results confirm the effectiveness of sensing of the refractive index, thickness, and incidence angle. The manuscript can be published in Sensors. There are some points for the authors to address.

1. About abbreviations: The “MS” for metastructure seems not necessary. The “GL” for graphene layer is also not necessary and could be confusing.

2. The authors mention that the JMS can be fabricated by etching. Could the authors give more detailed discussions?

3. Could the authors include more explanations about the graphene-based enhancement of PSHE to make the manuscript more explicit?

The English seems OK.

Author Response

The details can be seen in the attachment, which is a response letter to the comments of Reviewer.

Reviewer 3 Report

Review of manuscript entitled “Multiple physical quantities Janus metastructure sensor based on PSHE” by J. Sui et al.

My field of work is not related with theoretical developments, thus I cannot asses this manuscript. However, I give some comments.

1 Manuscript is well written and explains well the development of the work.

2 English description is good.

3 Fig. 1 shows well the phenomena and materials.

4 Plots in figures are clearly presented.

5 There are enough references.

6 Table 1 gives a good description of the calculated JMS parameters.

7 Table 2 compares well the theoretical characteristics of the proposed JMS with current or traditional sensors.

  8 Because this is a theoretical work you could give some guidelines to researchers that would like to develop experiments having the basis of your theoretical work.

Author Response

(The authors gave the same response as above.)

Round 2

Reviewer 1 Report

The authors took into account all my comments and therefore the work can be published